

# Coral reef fish assemblages exhibit signs of depletion in two protected areas from the eastern of Los Canarreos archipelago (Cuba, Caribbean Sea)

Zenaida María Navarro-Martínez[1], Maickel Armenteros[2], Leonardo Espinosa[3], Patricia González-Díaz[1] and Amy Apprill[4]

[1] Centro de Investigaciones Marinas, Universidad de La Habana, La Habana, Cuba
[2] Instituto de Ciencias del Mar y Limnología, Universidad Nacional Autónoma de México, Ciudad de México, México
[3] Empresa Nacional para la Protección de la Flora y la Fauna, La Habana, Cuba
[4] Marine Chemistry and Geochemistry Department, Woods Hole Oceanographic Institution, Woods Hole, MA, USA

Corresponding author
Zenaida María Navarro-Martínez,
zenamart21989@gmail.com

## ABSTRACT

Understanding the impact of marine protected areas on the distribution and composition of fishes is key to the protection and management of coral reef ecosystems, and especially for fish-based activities such as SCUBA diving and recreational fishing. The aim of this research is to compare the ichthyofauna structure in three areas in the eastern part of Los Canarreos archipelago in Cuba with different management schemes: Cayo Campos-Cayo Rosario Fauna Refuge (CCCR), Cayo Largo Ecological Reserve (CL) and non-protected area (nMPA), and considering habitat differences and depth variation. A total of 131 video transects were conducted using diver operated stereo-video (stereo-DOV) in November, 2015 in backreef and forereef along the CCCR, CL and the adjacent nMPA. We recorded 84 species and 27 functional groups suggesting high complementarity of functions. Several multispecies schools were observed along surveys, which explain the biomass peaks in some sites, mainly for Lutjanidae, Haemulidae and Carangidae. A concerning issue was the bare representation of critical functional groups and threatened species. The effect of sites nested within habitats was significant and the most important driver structuring fish assemblages, while MPA condition was not evident. Favorable habitat features (habitat heterogeneity and surrounding coastal ecosystems) are likely enhancing fish assemblages and counteracting the effects of pouching derived from insufficient management. We recommend immediate actions within a strategy of precautionary management including, but not limited to, the appointment of staff for the administration of CL, frequent monitoring and effective enforcement.

## INTRODUCTION

The designation of marine protected areas (MPAs) is a well-known approach to counteract the global deterioration of coastal ecosystems due to synergistic stressors, mainly fisheries and eutrophication (*Hughes et al., 2003*; *Hoegh-Guldberg & Bruno, 2010*; *Costanza et al., 2014*; *Harborne et al., 2017*). Indeed, successful MPAs are connected to the recovery of biological populations (*Roberts et al., 2001*; *Kelaher et al., 2014*; *Strain et al., 2019*). However, most MPA's interventions have suboptimal or poor results and a large fraction of MPAs worldwide suffer pouching and low levels of enforcement (*Mora et al., 2006*; *Costello & Ballantine, 2015*). Accordingly, MPA categories with different levels of fishing restrictions exist worldwide based on the criteria of the International Union for the Conservation of Nature (IUCN).

In Cuba, MPAs belonging to the National System of Protected Areas (SNAP) include a considerable portion of coastal ecosystems such as coral reefs (ca. 30%), seagrass beds (ca. 24%), and mangroves (ca. 35%) (*Perera-Valderrama et al., 2018*). These MPAs are classified after eight management categories, which have been adapted from IUCN categories (*Perera-Valderrama et al., 2018*). Two of these categories are relevant in our study: Ecological Reserve and Fauna Refuge. Ecological Reserve is equivalent to category II from the IUCN, and it protects important flora and fauna, as well as either complete or partial ecosystems, and can support nature-based tourism activities, such as low-impact hotels (*CNAP, 2013*; *Perera-Valderrama et al., 2018*). Fauna Refuge is equivalent to category IV from the IUCN. It conserves flora and fauna species and (or) populations, which can include management of habitats and ecosystems (*CNAP, 2013*; *Perera-Valderrama et al., 2018*). In theory, Ecological Reserve requires a stronger management (including fishing restrictions) and higher level of protection than Fauna Refuge; therefore, healthier fish assemblages should be expected in the former compared to the latter.

Not all Cuban MPAs have been equally studied, even despite some of them belong to the most restricted MPA categories (*Navarro-Martínez & Angulo-Valdés, 2015*). In this contribution, we focus on coral reef fish assemblages from two poorly studied MPAs in Los Canarreos archipelago (southwestern Cuba) (Fig. 1): Cayo Campos-Cayo Rosario Fauna Refuge (hereafter, CCCR) and Cayo Largo Ecological Reserve (hereafter, CL). Cayo Largo is a well-known tourist destination where recreational fisheries and SCUBA diving are carried out with success, depending partially on features of the fish assemblages. An integrative survey in Cayo Largo in 1998–1999 (*Alcolado, Claro-Madruga & Martínez-Daranas, 2001*) reported a significant deterioration of coral reef ecosystem. Posteriorly, marine biodiversity was studied in a coastal lagoon nearby to the Cayo Largo coral reef system (*Guardia, González-Sansón & Aguilar, 2003*), where human activities were likely not damaging the ecosystem.

Several broad-scale surveys in Cuban coral reefs included sites from Los Canarreos archipelago; namely: benthic communities (*Alcolado et al., 2010*, *2013*; *Caballero-Aragón et al., 2019*, *2020*), mesophotic biodiversity (*Reed et al., 2018*), and mesophotic fish assemblages (*Cobián-Rojas et al., 2021*). In all of these surveys, the ecological metrics in Los
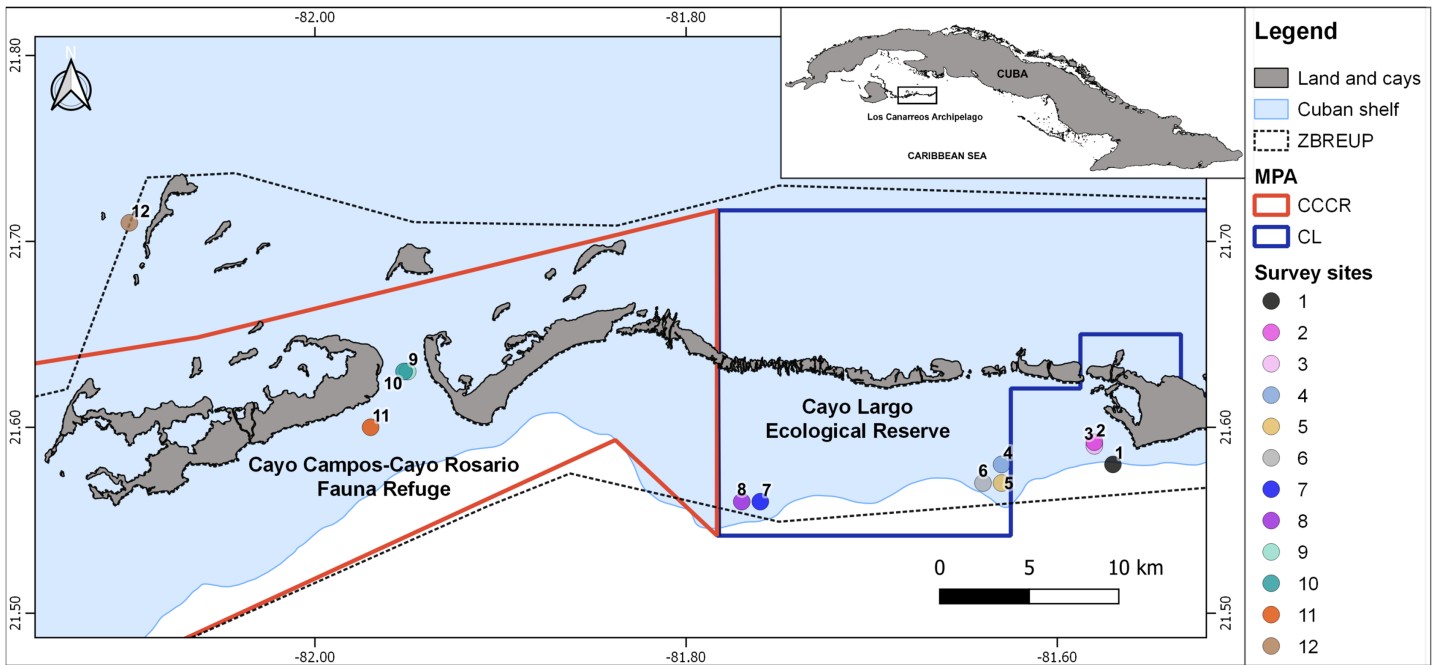

**Figure 1 Map showing the 12 survey sites in Los Canarreos archipelago.** Marine Protected Area (MPA), which are the Cayo Campos-Cayo Rosario Fauna Refuge (CCCR) and Cayo Largo Ecological Reserve (CL). ZBREUP. Zone under Special Regimen of Use and Protection (after *Ministry of Food Industry of Cuba (MINAL), 2012*). Survey sites were located in two reef habitats: backreef (sites: 2, 3, 4, 8, 9, 11) and forereef (sites: 1, 5, 6, 7, 10, and 12).

Canarreos's sites indicated rather regular conditions respect to other coral reef sites. However, no scientific information focused on the ichthyofauna in the MPAs CCCR and CL have been published.

Fishes represent excellent indicators of ecosystem health and MPA effectiveness, mainly in coral reefs (*Schmitter-Soto et al., 2017*; *Strain et al., 2019*; *Rojo, Anadón & García-Charton, 2021*). Functional traits of fishes are related to ecological processes in coral reefs such as secondary productivity, nutrient cycling, and trophic controls (*Bellwood et al., 2004*, *2019*; *Lefcheck et al., 2019*). Concordantly, reef fishes are likely one of the best studied assemblages in the Caribbean (*Miloslavich et al., 2010*), but given the increasing pressure due to fisheries and habitat deterioration, more studies are urgently needed.

The present work is aimed to compare the ichthyofauna structure in three areas in the eastern part of Los Canarreos archipelago with different management schemes: CCCR, CL, and non-protected area (nMPA), and considering habitat differences and depth variation. The biological data were collected during an expedition carried out along Los Canarreos archipelago in November 2015, as part of a comprehensive study targeted on fishes, benthic and microbial communities (*Weber et al., 2019*).

## MATERIALS AND METHODS

### Study zone

The study zone is located in the eastern part of Los Canarreos archipelago, southwestern Cuba, Caribbean Sea. The whole zone had been supposedly protected by a banning of

**Table 1 Sampling sites in Los Canarreos archipelago.** Sites are included in three management schemes based on the marine protected area (MPA) category: Cayo Largo Ecological Reserve (CL, MPA with high restriction, identified but non-legally approved), Cayo Campos-Cayo Rosario Fauna Refuge (CCCR, MPA with lower restriction, legally approved), and non-protected area (nMPA). N = number of transects.

| Site | Management scheme | Name of site or zone | Latitude (N) | Longitude (W) | Habitat type | Biotope | N | Average depth (m) |
|------|-------------------|----------------------|--------------|---------------|--------------|---------|---|-------------------|
| 1 | nMPA | Cueva del Negro | 21.58 | 81.57 | Forereef | Slope | 11 | 19.0 |
| 2 | nMPA | Punta Rabirrubia | 21.59 | 81.58 | Backreef | Crest | 12 | 3.7 |
| 3 | nMPA | Rabirrubia | 21.59 | 81.58 | Backreef | Patch reef | 10 | 3.4 |
| 4 | CL | Punta Barrera el Faro | 21.58 | 81.63 | Backreef | Patch reef | 10 | 3.2 |
| 5 | CL | Punta Barrera Ballenatos | 21.57 | 81.63 | Forereef | Spur and groove | 12 | 14.5 |
| 6 | CL | Acuario | 21.57 | 81.64 | Forereef | Slope | 12 | 12.4 |
| 7 | CL | Cabeza de la Estopa | 21.56 | 81.76 | Forereef | Terrace | 10 | 11.2 |
| 8 | CL | Cabeza de la Estopa | 21.56 | 81.77 | Backreef | Crest | 10 | 1.7 |
| 9 | CCCR | Quebrado del Rosario | 21.63 | 81.95 | Backreef | Crest | 10 | 2.5 |
| 10 | CCCR | Quebrado del Rosario | 21.63 | 81.95 | Forereef | Spur and groove | 12 | 11.5 |
| 11 | CCCR | Cayo Cantiles | 21.60 | 81.97 | Backreef | Crest | 11 | 3.4 |
| 12 | nMPA | Quebrado del Rosario | 21.71 | 82.10 | Forereef | Patch reef | 11 | 7.1 |

fisheries under a management regime termed Zone under Special Regimen of Use and Protection (ZBREUP, according to its name in Spanish). We surveyed two MPAs within the ZBREUP: Cayo Campos-Cayo Rosario Fauna Refuge (CCCR) and Cayo Largo Ecological Reserve (CL) (Fig. 1). CCCR was legally approved as Fauna Refuge since 2012 (*Consejo de Ministros, 2012*). CL is identified as an Ecological Reserve, but lack of any administration or enforcement despite of being included in a stricter category of management (*Perera-Valderrama et al., 2018*, *2020*). All the cays close to the study zone are uninhabited, but Cayo Largo hosts touristic infrastructure that includes one airport, one marina and several hotels. The zone includes typical tropical marine habitats such as sandy beaches, mangroves, seagrass beds and coral reefs. Coral reefs include several biotopes along the study area: patch reefs, reef crest, terrace, slope, and spur and groove.

Field surveys were conducted under the permission No. 2015/25 for accessing to natural and mountainous areas, emitted by the Ministerio de Ciencia, Tecnología y Medio Ambiente de Cuba (CITMA), in favor to the Centro de Investigaciones Marinas, Universidad de La Habana.

## Sampling and video analysis

A scientific expedition was carried out from November 25[th] to 30[th], 2015 for surveying fish assemblages. Three survey sites were located inside the CCCR MPA, five sites inside the CL MPA, and four sites outside of MPA (hereafter nMPA) (Fig. 1, Table 1). Surveys were done in two types of coral reef habitats: backreef which included patch reef and crest biotopes (1–5 m depth), and forereef which included deeper patch reef, terrace, slope, and spur and groove biotopes (7–22 m depth) (Table 1, Fig. S1). The heterogeneity of the coral reefs in the zone and logistic restrictions did not allow a balanced sampling of sites covering all the combinations of factors.

The sampling was done using diver operated stereo-video (stereo-DOV) technique. The stereo-DOV equipment consisted on a Canon VIXIA HF S21 Full HD camcorder with a Raynox HD 6600 Pro conversion lens. The stereo-DOV equipment was calibrated, which allowed to record accurate and precise measurements. Stereo-DOV system, and the calibration hardware and software were provided by SeaGIS Pty Ltd.

Fish survey was conducted by two SCUBA divers, one of them operated the stereo-DOV equipment and the other held a metric tape (attached to the camera system) to fix the starting and ending point of each transect. Divers were separated ca. 35 m to avoid disturbance in the area and they kept communication by tractions in the tape. The average speed of swimming was 0.3 m s$^{-1}$, the tilt of the camera was 30° and the distance respect to the bottom was approximately 0.5 m. The sampling unit was a transect of 25 m long and 5 m width, covering an area of 125 m$^2$. Whenever possible, 12 transects in a straight line were made by site with 10 m apart each other (Table 1). However, some sites had less than 12 transects (10 or 11) because limits imposed by the SCUBA diving (*e.g.*, diving time).

A total of 131 video transects were analyzed in the laboratory, using the software EventMeasure, version 3.32. (*SeaGIS, 2011*). All fishes inside transects were identified, counted and recorded the fork length, when possible. Because of the limitations inherent to the used technique (*Navarro-Martínez et al., 2017*; *Goetze et al., 2019*), we were unable to observe very cryptic or tiny species (*e.g.*, gobies, flounders, blennies). Therefore, we excluded from the dataset all the fishes not identified at species or genus level.

Fish fork length was recorded to the nearest millimeter. We accepted those measurements with precision values lower than 10 mm and residual mean square (RMS) error lower than 20 mm, as recommended for stereo-DOV (*Goetze et al., 2019*). The precision to length ratio was lower than 5%. Considering the species included in the biomass analysis, the average value of precision along the entire survey was 3.8 mm and RMS error was 9.2 mm. The size range was: 58.1–625.1 mm, which correspond to one specimen of *Scarus* sp. and *Epinephelus striatus*, respectively. Precision, RMS error, and precision to length ratio are metrics provided by the software for each measurement. The body weight (W, in grams) was calculated using the length-weight allometric relation recorded per species:

$$W = a \times L^b$$

where $L$ is the fork length (in centimeters), and $a$ and $b$ are species-specific parameters calculated for the geographic region (*Claro, Lindeman & Parenti, 2001*; *Froese & Pauly, 2020*).

Water depth was calculated per transect by averaging four depth records taken at distances of 0, 8, 16, and 24 m along each transect, based on a diving computer attached on the stereo-DOV system.

## Data analyses

A matrix of fish species abundance (individuals 125 m$^{-2}$) × samples was constructed for the estimation of species density and abundance. A second matrix of abundance was built with a subset of those species classified as threatened or near threatened, hereafter

threatened species, which included four IUCN categories: critically endangered, endangered, vulnerable, and near threatened (*IUCN, 2022*). A third matrix of biomass was built by multiplying body weight by abundance (kg 125 m$^{-2}$); this matrix only included the five commercially and ecologically relevant families Carangidae, Haemulidae, Lutjanidae, Scaridae, and Serranidae.

A fourth matrix of functional group abundance × samples was constructed. Fish species were assigned to functional groups based on the combination of three traits: mobility, trophic guild, and body size, which followed the criteria of *Micheli et al. (2014)* and FishBase (*Froese & Pauly, 2020*). The mobility categories were: cryptic, roving, midwater, and pelagic. Trophic guilds were based on nine categories: (i) browsers, (ii) grazers (both of them herbivores), (iii) planktivores, (iv) macroinvertivores (invertebrate feeders, of 25–50 cm maximum length), (v) microinvertivores (small-invertebrate feeders of <30 cm length), (vi) sessile invertivores (sessile-invertebrate feeders), (vii) predators (invertebrate feeders and piscivores), (viii) piscivores, and (ix) omnivores. Body size categories were based on maximum body length range: <25, 25–50, 50–100, and >100 cm. From the matrix, we estimated the functional group richness and functional redundancy of fishes, also based on the criteria of *Micheli et al. (2014)*. Functional group richness was defined as the number of functional groups. Functional redundancy was defined as the number of species per functional group, which was averaged by transect.

All variables were tested using a permutational analysis of variance (PERMANOVA, *Anderson, Gorley & Clarke, 2008*). We built a mixed model with three factors and one covariate: MPA category (three levels: CL, CCCR, nMPA; fixed effect) crossed with HABITAT (two levels: Forereef, Backreef; fixed effect), and SITE nested within HABITAT (12 levels: sites 1–12; random effect). DEPTH was set as covariate and transects were used as replicates. We selected 9,999 permutations of residuals under a reduced model. Sequential sum of squares (type I) was used to order terms in the model according to the objectives and hypotheses to be tested: MPA category, HABITAT, DEPTH, MPA effect × HABITAT, and SITE (HABITAT). Mean, standard error (SE) and 0.95 confidence interval (CI) were calculated according to the statistical design described above; *i.e.*, averaging transects per HABITAT level, and MPA level. We tested differences according to this statistical design for six variables: (1) abundance, (2) species density, (3) functional group richness, (4) functional redundancy, (5) biomass of selected families, and (6) abundance of threatened species. PERMANOVA was applied with the following settings: Euclidian Distance as resemblance measure, 9,999 permutations, sums of squares type I (sequential), and permutation of residuals under a reduced model.

We tested differences in the multivariate structure of the fish assemblages using the matrix of species abundance. Abundance values were square root transformed to reduce the influence of the most abundant species on the multivariate pattern. A PERMANOVA test was applied using Bray-Curtis similarity index as resemblance measure, the other settings were the same as for univariate tests. The similarity pattern of the fish assemblages across the samples was represented through a non-metric multidimensional scaling (NMDS). The Bray-Curtis similarity index was used as the resemblance measure with 500

starting configurations. For better visualization, the original matrix of species abundance × sample was averaged by site to obtain a matrix of mean species abundance × site.

R language was used for data processing, descriptive statistics, and graphs (*R Core Team, 2020*). Graphs were created using the ggplot2 (*Wickham, 2016*) and Hmisc (*Harrell, 2021*) packages. The PERMANOVA test and multivariate analyses were done in the software PRIMER 7.0.21 (*Clarke et al., 2014*).

## RESULTS

### Taxonomic structure

Over the 12 reef sites differing in MPA status and habitat type, we recorded 15,059 fishes, of which 14,794 (98%) were identified mostly to species level. We reported 80 species, three genera (*Calamus*, *Kyphosus*, and *Pterois*), and the species complex *Acanthurus bahianus/Acanthurus tractus* (Table S1). For the quantitative analyses, we used the genera and the species complex as equivalent to species resulting in 84 entries. The species belong to 24 families, with Haemulidae, Pomacentridae, Labridae, Acanthuridae, Scaridae, and Lutjanidae the best represented with more than 1,000 individuals and seven species (excepting Acanthuridae). The most abundant species were (in brackets the number of individuals): *Haemulon sciurus* (2,230), *Thalassoma bifasciatum* (1,613), *Acanthurus coeruleus* (1,291), and *Haemulon flavolineatum* (1,004).

Fish abundance varied significantly between sites with a mean of 113 ind. 125 m$^{-2}$ (range: 18–717 ind. 125 m$^{-2}$) (Fig. 2A). However, no differences were detected between MPA category or between habitats (Table 2). Species density varied significantly between sites with a mean of 16 species 125 m$^{-2}$ (range: 5–30 species 125 m$^{-2}$). However, habitat type explained an amount of variance (19%) similar to site (21%) suggesting that it is an important source of variation as well (Table 2). Actually, species density tends to be higher in backreef compared to forereef (Fig. 2B).

We detected ten species which are classified in four threatened categories according to *IUCN (2022)*: *Epinephelus striatus* as critically endangered; *Aetobatus narinari* as endangered; *Ginglymostoma cirratum*, *Lachnolaimus maximus* and *Lutjanus cyanopterus* as vulnerable; *Balistes vetula*, *Hypanus americanus*, *Lutjanus analis*, *Mycteroperca venenosa*, and *Scarus guacamaia* as near threatened. The abundance of threatened species was significantly different only between MPA categories (Table 2), with the least abundance located in the CL-backreef (Fig. 3A). The most abundant threatened species were *Lachnolaimus maximus*, *Balistes vetula*, and *Epinephelus striatus* with averaged abundance of 0.24, 0.11 and 0.08 ind. 125 m$^{-2}$, respectively. Backreefs in Cayo Largo had only one threatened species with very low abundance, conversely forereefs in nMPA had six threatened species with higher abundance (Fig. 3B).

Biomass was significantly variable between sites, and no other clear trends emerged between MPA categories or between habitat types (Fig. 4A). The families with higher biomass were Lutjanidae and Haemulidae, followed by Scaridae (Fig. 4B).

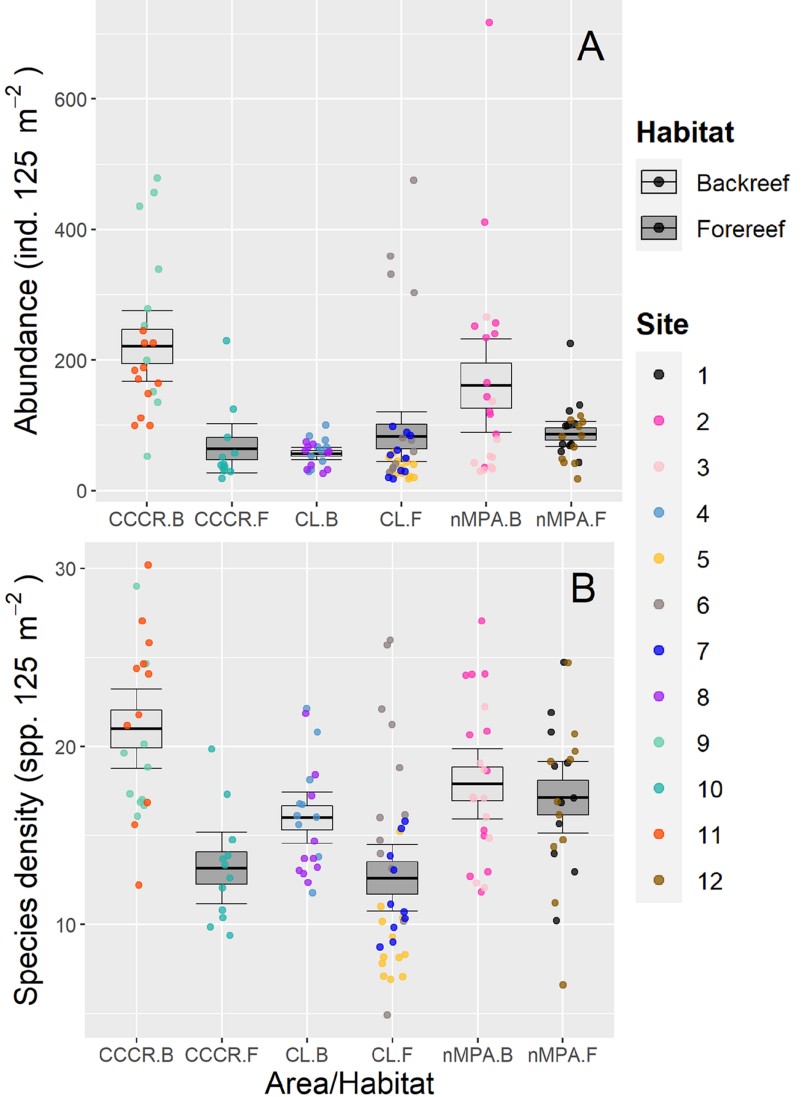

**Figure 2 (A) Abundance and (B) species density of reef fishes recorded along Los Canarreos archipelago in Cayo Campos-Cayo Rosario Fauna Refuge (CCCR), Cayo Largo Ecological Reserve (CL), and non-protected sites (nMPA).** Horizontal lines indicate the mean, boxes indicate the standard error, and error lines indicate the 0.95 confidence interval. Each point represents a transect (125 m$^2$) carried out in different sites (1–12) and grouped by reef habitats: Backreef (B) and Forereef (F).

## Functional structure

Twenty-seven functional groups were observed during the study (Table S1, Fig. S2). Most redundant functional groups, according to the number of species (in brackets), were the roving browsers (six), roving grazers (six), roving predators (eight), and roving macroinvertivores (15). Conversely, large roving browsers, small cryptic macroinvertivores, pelagic macroinvertivores, midwater macroinvertivores, roving sand macroinvertivores, cryptic macroinvertivores, and pelagic planktivores were scarcely represented by one species observed in just one site. The most abundant functional groups

**Table 2 Results of the permutational analysis of variance (PERMANOVA) on ecological variables of fish assemblages from Los Canarreos archipelago.** Mixed model with three factors and one covariate: MPA category (three levels; fixed effect) crossed with HABITAT (two levels; fixed effect), and SITE nested within HABITAT (12 levels; random effect). DEPTH was set as covariate and transects were used as replicates.

| Variable | Source of variation | Degrees of freedom | Pseudo-F | p-value | Component of variation (%) |
|---|---|---|---|---|---|
| Abundance | MPA | 2 | 1.55 | 0.28 | 11 |
| | HABITAT | 1 | 1.60 | 0.25 | 10 |
| | DEPTH | 1 | 0.18 | 0.68 | 0 |
| | MPA × HABITAT | 2 | 1.43 | 0.30 | 14 |
| | SITE (HABITAT) | 6 | 5.86 | **0.0001** | 27 |
| | Residual | 118 | | | 38 |
| Species density | MPA | 2 | 2.78 | 0.13 | 16 |
| | HABITAT | 1 | 4.42 | 0.07 | 19 |
| | DEPTH | 1 | 0.10 | 0.76 | 0 |
| | MPA × HABITAT | 2 | 1.24 | 0.35 | 9 |
| | SITE (HABITAT) | 6 | 4.44 | **0.0006** | 21 |
| | Residual | 118 | | | 36 |
| Threatened species | MPA | 2 | 6.11 | **0.03** | 14 |
| | HABITAT | 1 | 1.49 | 0.25 | 4 |
| | DEPTH | 1 | 3.50 | 0.09 | 14 |
| | MPA × HABITAT | 2 | 3.51 | 0.09 | 14 |
| | SITE (HABITAT) | 6 | 0.47 | 0.83 | 0 |
| | Residual | 118 | | | 55 |
| Biomass | MPA | 2 | 1.68 | 0.28 | 13 |
| | HABITAT | 1 | 1.38 | 0.29 | 8 |
| | DEPTH | 1 | 0.16 | 0.68 | 0 |
| | MPA × HABITAT | 2 | 0.64 | 0.61 | 0 |
| | SITE (HABITAT) | 6 | 2.86 | **0.02** | 24 |
| | Residual | 118 | | | 54 |
| Functional group richness | MPA | 2 | 1.14 | 0.39 | 6 |
| | HABITAT | 1 | 0.91 | 0.37 | 0 |
| | DEPTH | 1 | 1.88 | 0.19 | 18 |
| | MPA × HABITAT | 2 | 1.11 | 0.40 | 8 |
| | SITE (HABITAT) | 6 | 5.82 | **0.0001** | 28 |
| | Residual | 118 | | | 40 |
| Functional redundancy | MPA | 2 | 2.82 | 0.13 | 17 |
| | HABITAT | 1 | 4.54 | 0.08 | 19 |
| | DEPTH | 1 | 0.11 | 0.73 | 0 |
| | MPA × HABITAT | 2 | 1.24 | 0.36 | 9 |
| | SITE (HABITAT) | 6 | 4.32 | **0.0005** | 21 |
| | Residual | 118 | | | 35 |
| Multivariate structure | MPA | 2 | 2.55 | **0.0016** | 13 |
| | HABITAT | 1 | 5.84 | **0.0002** | 19 |
| | DEPTH | 1 | 1.10 | 0.37 | 4 |
| | MPA × HABITAT | 2 | 1.79 | **0.024** | 14 |
| | SITE (HABITAT) | 6 | 4.26 | **0.0001** | 18 |
| | Residual | 118 | | | 31 |

**Note:**
Significant values at $p < 0.05$ are in bold.

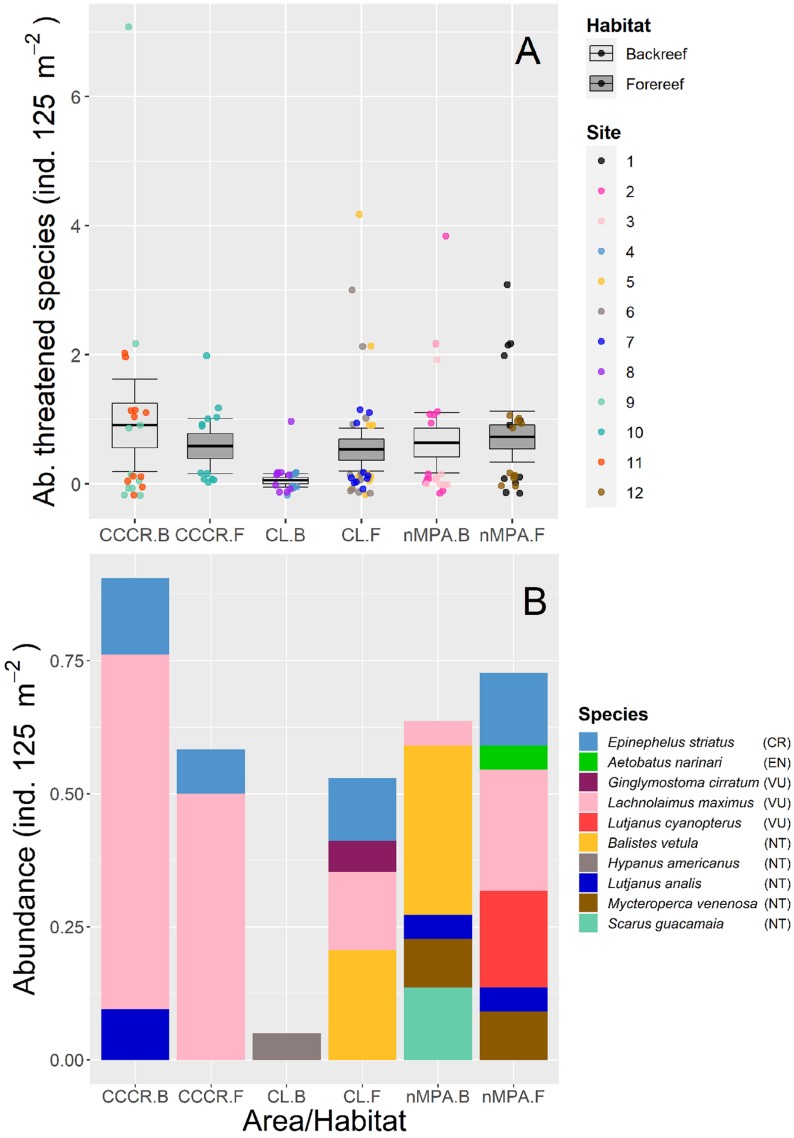

**Figure 3 (A) Total abundance and (B) stacked abundance per species of threatened fishes in Los Canarreos archipelago: in Cayo Campos-Cayo Rosario Fauna Refuge (CCCR), Cayo Largo Ecological Reserve (CL), and non-protected sites (nMPA), and grouped by reef habitats: Backreef (B) and Forereef (F).** Horizontal lines indicate the mean, boxes indicate the standard error, and error lines indicate the 0.95 confidence interval. Each point represents a transect (125 m²) carried out in different sites (1–12). The ten threatened/near threatened fish species were classified by the IUCN as critically endangered (CR), endangered (EN), vulnerable (VU), and near threatened (NT).

were the roving macroinvertivores followed by the roving grazers, cryptic grazers and roving planktivores (Fig. S2).

Functional group richness varied significantly between sites (Table 2), with a mean of nine functional groups (range: 4–14 functional groups) (Fig. 5A). Functional redundancy also varied significantly between sites with a mean of 0.59 species per functional group (range: 0.19–1.07 species per functional group). Site explained 21% of the total variance,

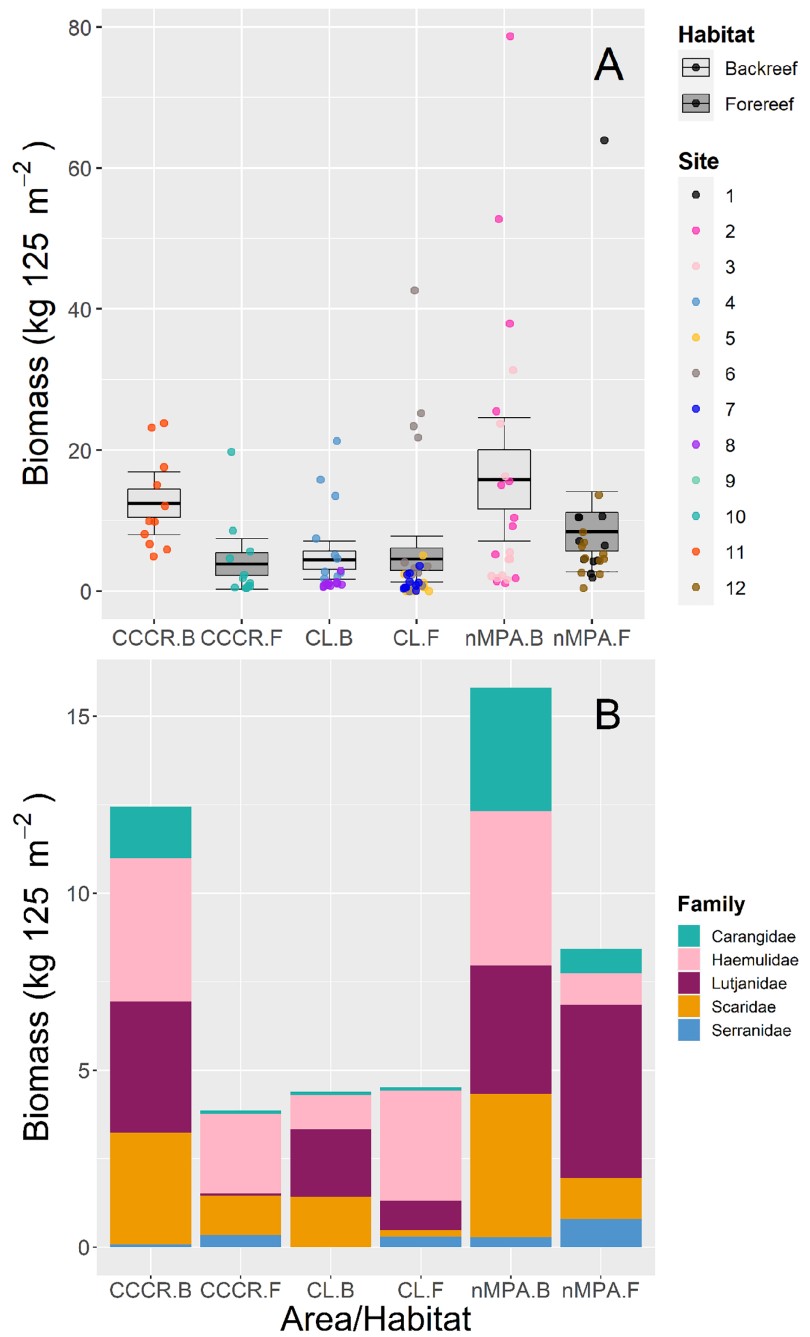

**Figure 4** (A) Total biomass and (B) stacked biomass per family of five fish families ecologically and/or commercially important in Los Canarreos archipelago in Cayo Campos-Cayo Rosario Fauna Refuge (CCCR), Cayo Largo Ecological Reserve (CL), and non-protected sites (nMPA), grouped by reef habitats: Backreef (B) and Forereef (F). Horizontal lines indicate the mean, boxes indicate the standard error, and error lines indicate the 0.95 confidence interval. Each point represents a transect (125 m$^2$) carried out in different sites (1–12).

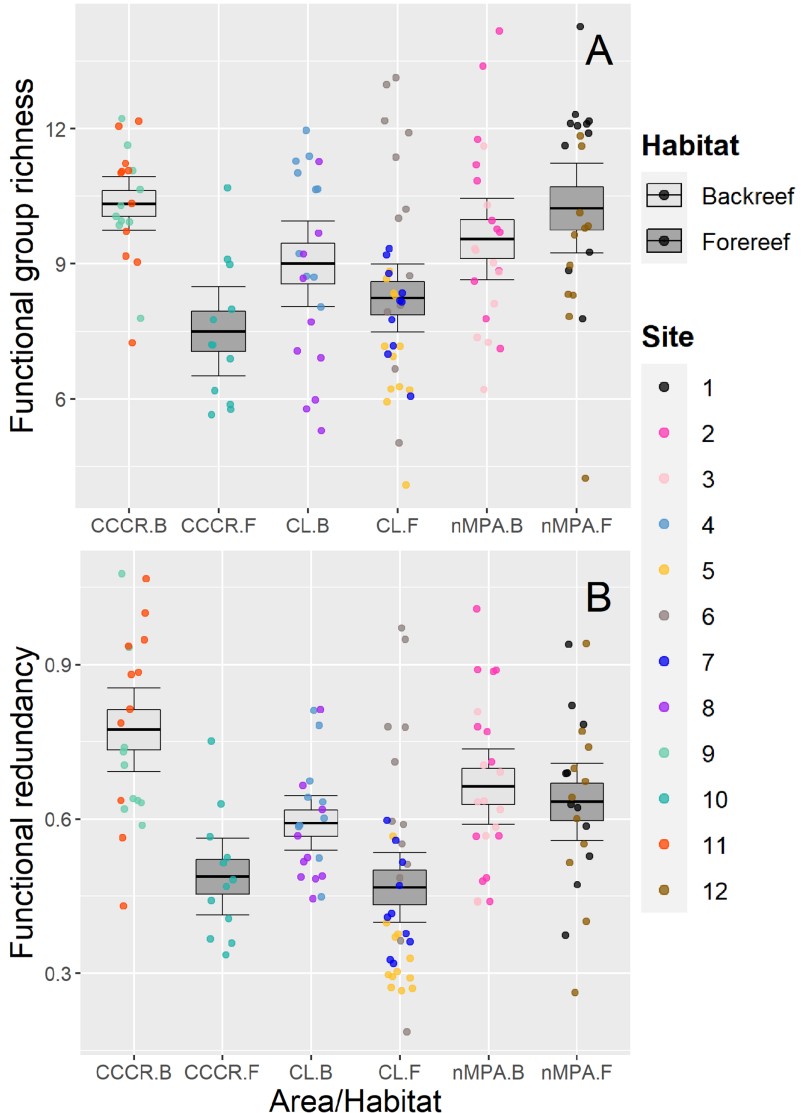

**Figure 5 (A) Functional group richness and (B) functional redundancy of reef fishes recorded along Los Canarreos archipelago in Cayo Campos-Cayo Rosario Fauna Refuge (CCCR), Cayo Largo Ecological Reserve (CL), and non-protected sites (nMPA).** Functional group richness is given as number of functional groups per transect, and functional redundancy is given as average number of species per functional groups along each transect. Horizontal lines indicate the mean, boxes indicate the standard error, and error lines indicate the 0.95 confidence interval. Each point represents a transect (125 m²) carried out in different sites (1–12) and grouped by reef habitats: Backreef (B) and Forereef (F).

similar to habitat (19%) and MPA (17%), suggesting that all of them influenced the functional redundancy (Table 2). Actually, redundancy tended to be higher in backreefs, but only within MPAs (*i.e.*, in CL and CCCR) (Fig. 5B).

## Multivariate structure

The multivariate structure varied significantly after several factors, listed according to the amount of explained variance: Habitat, site, interaction MPA × habitat, and MPA category

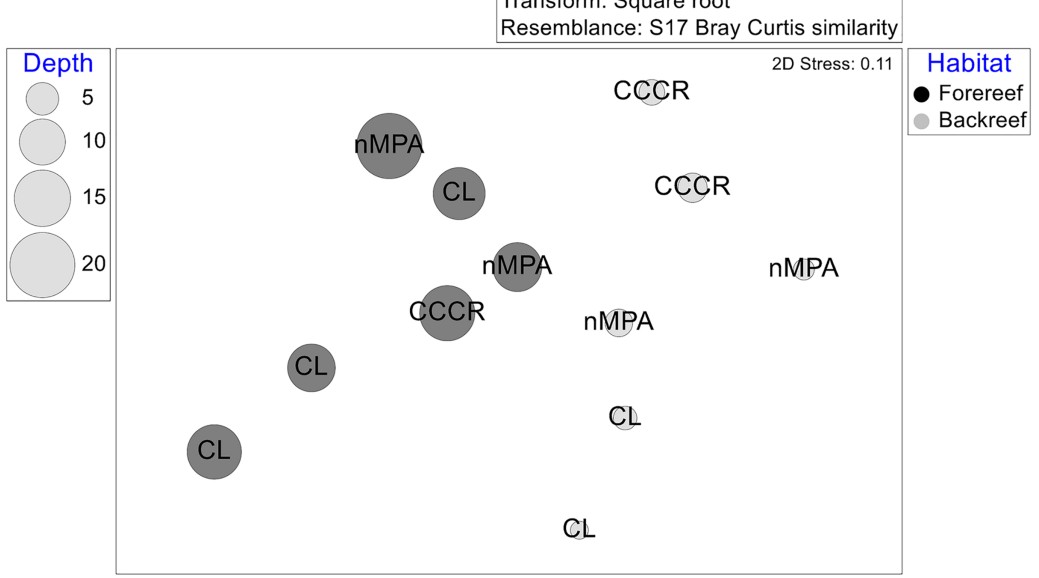

**Figure 6 Multivariate structure of the fish assemblages in 12 sites represented in an ordination by a non-metric multidimensional scaling based on transformed averaged abundance per site.** Labels correspond to MPA categories: Cayo Campos-Cayo Rosario Fauna Refuge (CCCR), Cayo Largo Ecological Reserve (CL), and non-protected sites (nMPA). Symbols correspond to habitat type: Forereef (F), Backreef (B). Size of bubbles indicates the averaged depth (m) per site.

(Table 2). The ordination by NMDS showed that habitat (and some extension depth) causes some grouping of sites (Fig. 6).

## DISCUSSION

This research contributes to the ichthyofauna knowledge from a poorly explored coral reef region of the Caribbean Sea, namely the eastern part of Los Canarreos archipelago. We provide, for first time for this region, published comprehensive information regarding the structure of coral reef fish assemblages from a taxonomic (abundance, density and biomass) and functional (richness and redundancy) perspectives, and how these features relate to MPA category, habitat type and depth. We acknowledge that our study lacks temporal replication (*i.e.*, a single expedition) that limits the reach of the inference. Nevertheless, our study still provides new insights into the effect of MPA condition to the reef system, and the effects of ecological drivers on the fish assemblages.

We reported that MPA's condition did not have a significant effect over the fish assemblage structure and ecological metrics likely because the poor management. Although CCCR was approved as MPA since 2012 (*Consejo de Ministros, 2012*), it confronts, as other Cuban MPAs, difficulties in enforcement, mainly due to limited resources and conditions (*Perera-Valderrama et al., 2018*). In addition, CL lacked an administration staff and enforcement.

In this regard, only the abundance of threatened fish species and the multivariate structure of fishes, showed significant differences based on MPA status. Threatened species are those whose population size and geographic range have been reduced, usually as result

of habitat deterioration and overfishing (*IUCN, 2012*); and they are useful indicators of protection effectiveness in MPAs and habitat availability (*Navarro-Martínez et al., 2022*). However, despite the significant MPA effect, threatened species were scarcely represented in most of our survey sites. Paradoxically, the abundance of these species was the lowest within CL, which is in accordance with the lack of enforcement already mentioned, at the time that represents a concern; since this MPA was identified due to its natural resources compatible with the touristic activities developed there.

Two key metrics of fish assemblages frequently used to evaluate the effectiveness of MPAs are the biomass, and the occurrence of particular functional groups (*e.g.*, large piscivores and predators). Both of these metrics were consistently similar (*i.e.*, not significantly different) between MPA categories. The biomass of ecologically and commercially important families showed a contrasting panorama along survey sites. For instance, we found very low biomass values in some survey sites (mostly in CL; generally <2 kg 125 m$^{-2}$). However, several multispecies schools (Lutjanidae, Haemulidae and Carangidae) were observed along other survey sites resulting in high biomass (>12 kg 125 m$^{-2}$) in both protected and non-protected sites. The fish families Lutjanidae, Haemulidae, Carangidae, Serranidae, and Scaridae usually show scarce biomass in deteriorated coral reefs where fisheries are allowed or MPA's enforcement is ineffective (*Valdivia, Cox & Bruno, 2017*, *Duran et al., 2018*, *Navarro-Martínez et al., 2022*). This fact responds to their ecological role and to their historical extraction in non-protected areas through fisheries, including Cuban waters (*Baisre, 2018*). Therefore, these fish families are important proxies of healthy/protected coral reefs. In this context, very low density and biomass of herbivorous (0.02–0.03 ind. m$^{-2}$; 2.4–3.3 g m$^{-2}$, respectively) and carnivorous (includes to Lutjanidae and Serranidae: 0.002–0.004 ind. m$^{-2}$; 0.02–0.8 g m$^{-2}$, respectively) fishes were recorded in surveys done in Cayo Largo reef crest in 2012 (*Alcolado et al., 2013*). Our records of biomass for these groups were higher near habitats/sites in CL and nMPA backreefs (for comparisons, we converted our values from kg 125 m$^{-2}$ to g m$^{-2}$, with the following result, Scaridae: 11.36 g m$^{-2}$, 32.41 g m$^{-2}$; Serranidae: 0.04 g m$^{-2}$, 2.30 g m$^{-2}$, Lutjanidae: 15.24 g m$^{-2}$, 28.96 g m$^{-2}$, respectively), but direct comparisons should be avoided since both studies included different survey techniques and survey sites. On the other hand, our current results of biomass for ecologically and commercially important families are comparable but mostly lower than other records from Cuba and the Greater Caribbean (Table 3).

Several functional groups were scarcely represented along the study region; in particular, larger individuals of Serranidae and Scaridae. These families are frequently evaluated in studies of MPA effectiveness due to their sensibility to anthropogenic impact and their importance in the ecosystem functioning (*Valdivia, Cox & Bruno, 2017*; *Lefcheck et al., 2019*). Larger groupers were barely represented by only four species with low abundance: two piscivore species of *Mycteroperca* and two predator species of *Epinephelus* genera. Similarly, larger browser parrotfishes (*i.e.*, herbivores) were barely represented in terms of abundance.

Our results suggest that the ichthyofauna from CCCR and CL MPAs exhibit signs of depletion such as low biomass and loss of key groups. This occurs despite that they have

**Table 3 Comparison of reports of biomass (g m$^{-2}$) for the families Lutjanidae, Serranidae and Scaridae in different locations from the Greater Caribbean and the eastern reefs of Los Canarreos archipelago, Cuba.** Cayo Campos-Cayo Rosario Fauna Refuge (CCCR), Cayo Largo Ecological Reserve (CL) and non-protected area (nMPA).

| Area (reference) | Habitat (depth) | Fish group | Protected/fully protected sites | General use areas | Unprotected sites |
|---|---|---|---|---|---|
| Eastern reef of Los Canarreos archipelago, Cuba (This study) | Forereef habitat (7–22 m) | | CCCR | CL | nMPA |
| | | Lutjanidae | 0.56 | 6.66 | 39.16 |
| | | Serranidae | 2.69 | 2.41 | 6.37 |
| | | Scaridae | 8.92 | 1.47 | 9.29 |
| Jardines de la Reina, Cuba (Navarro-Martínez et al., 2022) | Terrace/Reef slope/Spur and groove (7–24 m) | Lutjanidae | 27.7 | | 11.0 |
| | | Serranidae | 10.8 | | 5.0 |
| | | Scaridae | 15.3 | | 9.2 |
| Belize (Cox et al., 2017) | Spur and groove (15–18 m) | Lutjanidae | 6.9 | 3.1 | 10.5 |
| | | Serranidae | 8.0 | 4.3 | 4.9 |
| | | Scaridae | 29.1 | 34.0 | 32.2 |
| Greater Caribbean (Vallès & Oxenford, 2014)* | Forereef habitat (undefined depth) | Lutjanidae | 8.2 | | 10.1 |
| | | Serranidae | 3.7 | | 2.8 |
| | | Scaridae | 19.8 | | 16.3 |
| Caribbean (Jackson et al., 2014) | Coral reef habitats (<120 m) | Scaridae | | 13.5 (Range: 0.7–60.7) | |
| Southwater Caye Belize (Mumby et al., 2021) | Complex *Orbicella* reef (10–12 m)** | Lutjanidae | ~4, 10 | | ~4 |

**Notes:**
* Greater Caribbean (including Bahamas, Netherland Antilles, Cayman, St. Vincent, Turks and Caicos, Belize, Venezuela, Costa Rica, Virgin Islands, Mexico, Jamaica, Cuba, Panama, USA, Puerto Rico, Nicaragua, Dominican Republic).
** Only surveys data from 2018 were included.

been protected from fisheries since the ZBREUP's creation; that is, since 1997 in CL only (*MIP, 1997*) and since 2012 (*Ministry of Food Industry of Cuba (MINAL), 2012*) for all the study area (Fig. 1). In principle, coral reefs closer to the touristic facilities at Cayo Largo should suffer the highest impact due to pollution (chemical and acoustic), physical disturbance and pouching, while the remoteness of the other sites (*e.g.*, CCCR and non-MPA 12) hampers the characterization of anthropogenic impacts and illegal activities, which in turn limits the management. For instance, poor management signals have been reported in other studies in Los Canarreos, which reported illegal fishing in Cayo Largo and their proximities (*Alcolado, Claro-Madruga & Martínez-Daranas, 2001*; *Guardia & González-Díaz, 2002*; *Azanza et al., 2018*). Pouching explains the lack of larger sized fishes and depletion of their abundance in most of our study sites. This result suggests the low representativeness of critical functions to the coral reef ecosystem such as herbivory and transference of energy to top trophic levels.

Importantly, synergetic factors are likely affecting different levels of trophic webs in this area of Los Canarreos. For instance, high density of macroalgae coverage was reported as early as 1998–1999 by *Alcolado, Claro-Madruga & Martínez-Daranas (2001)* likely because of the low density of herbivores. Recently, a comparison between coral reefs from Florida Keys (USA), Jardines de la Reina (Cuba) as well as the same sites we surveyed in
this study, reported comparably high algae and low coral coverage along Los Canarreos (*Weber et al., 2019*). Similarly, *Caballero-Aragón et al. (2019)* found rather medium-low estimates of coral metrics (diversity and living coral cover) and high mortality in Los Canarreos, compared with other Cuban reefs. Unfortunately, these results fit into the general trend of coral reef deterioration worldwide (*Hughes et al., 2017*).

On the other hand, fish assemblages of the region possess some favorable features typical of healthy habitats. For instance, assemblages were taxonomically and functionally diverse with 84 species and 27 functional groups, indicating high diversity and complementarity of functions that are associated with healthy coral reefs (often within successful MPAs) (*Bellwood et al., 2004*, *2019*; *Micheli et al., 2014*). In addition, a few sites showed higher biomass of important families (*e.g.*, Lutjanidae, Carangidae, Haemulidae) than others from the Greater Caribbean (*Vallès & Oxenford, 2014*) and similar biomass as in the successful Jardines de la Reina National Park (*Navarro-Martínez et al., 2022*) (Table 3). Nevertheless, considering the rather deficient management history from these areas (see above), such "high" diversity features are likely remnants of the coral reef fish assemblages that used to inhabit the region (*Jackson, 1997*).

In this context, the current habitat heterogeneity is playing a critical role preserving and structuring the reef fish assemblages. While MPA category and depth lacked significant effects for most of the tested metrics, our results suggest an important effect of sites (nested within habitats) on the fish assemblage structure. The significant effect of sites points to the important role of local environmental features on ichthyofauna, which are often elusive to identify (*e.g.*, bottom rugosity, enhanced primary production). Therefore, in addition to our quantitative data about fish assemblage structure, we analyzed the underwater footage for reef characteristics and the geomorphology of the archipelago to determine the influence of cays or other habitats. We think that at least two ecological drivers may be important in shaping the assemblage structure in the region: habitat heterogeneity and proximity of coastal habitats. These natural drivers may counteract in some degree the lack of effectiveness of the MPAs and they will be discussed in more details below; however, improvements in MPA management are still essential for raising better ichthyofauna conditions (*e.g.*, more abundant threatened species, higher biomass of critical functional groups).

Habitat heterogeneity is notably high in Los Canarreos region. In our study we recorded six different biotopes from 12 sites: crest, patch reef (shallow and deep), terrace, slope, and spur and groove. *Caballero-Aragón et al. (2019)* reported some of the largest values of reef rugosity (a measure of bottom complexity) in backreefs and forereefs of Los Canarreos compared to other sites around the Cuban archipelago. This heterogeneity promotes shelter and trophic resources for fishes (*Gratwicke & Speight, 2005*). Los Canarreos coral reefs are very close to other relevant coastal ecosystems, mainly mangroves and seagrass beds. The complex of interconnected mangrove-seagrass bed-coral reef ecosystems provides nursery areas that in turn enhance fish assemblages (*Nagelkerken et al., 2002*; *Adams et al., 2006*; *Serafy et al., 2015*). This may explain the relatively high taxonomic and functional diversity of fishes in the region. Even more, spawning aggregation sites occur for

the snapper species *Lutjanus griseus*, *Lutjanus analis*, and *Lutjanus cyanopterus* in Cayo Avalos (*Claro & Lindeman, 2003*), which is relatively close to the study area.

Additionally, the proximity to many vegetated cays and the Zapata swamp (the largest swamp in the insular Caribbean) likely enhances the delivery of nutrients supporting high levels of primary production in the coral reefs. A recent study including 73 reefs around the Cuban archipelago (H. Caballero-Aragón, 2002, 2013, unpublished data) reports the highest concentration of particulate organic carbon and chlorophyll in the south of Zapata Swamp, relatively close to Los Canarreos. Other studies also reported nutrient enrichment in the coral reef system of Los Canarreos (*Alcolado, Claro-Madruga & Martínez-Daranas, 2001*; *Reed et al., 2018*; *Weber et al., 2019*). But important knowledge gaps remain for explaining inter-site variability of fish assemblages, which in turn weaken the success of management strategies. Examples of relevant knowledge gaps are the quantification of the habitat complexity and the effects of runoff from Zapata Swamp.

Our results suggest a need to improve the management/enforcement of Los Canarreos MPAs. All of the study area appears as protected/regulated, at least on paper. This "in-paper management" hides the current situation of this supposedly protected region and offer wide possibilities of illegal activities beyond of poaching, such as extraction of ornamental organisms such as black corals and some mollusk species. The fact that recorded sites are relatively far from human settlement represents a clear advantage for the MPA management, considering that human settlement tends to negatively affect the marine biodiversity (*Mora, 2008*). Therefore, the synergistic effects of natural and adequate management facts may preserve and even improve the ichthyofauna structure, the habitat condition, but also other processes that occur in this area, *e.g.*, turtles nesting (*Azanza et al., 2018*). In addition, current touristic activities carried out in the area (SCUBA diving, and catch and release fisheries) could additionally benefit, since they depend on specific conditions including habitat mosaic and coral reef fishes.

## CONCLUSIONS

The CCCR and CL MPAs lack significantly healthier coral reef fishes when compare to non-protected sites raising doubts about their effectiveness for conservation. Low abundance of threatened species and key functional groups, as well as low biomass, highlight the necessity of stronger management/enforcement in both MPAs. The current high taxonomic and functional diversities, and the occurrence of multispecies schools are likely remnants of healthy reef fish assemblages that used to inhabit the once pristine coral reefs of the region. The overarching effect of local environment (*i.e.*, sites) points to the important role of habitat heterogeneity on fish assemblage structure. Favorable habitat features are likely enhancing fish assemblages and counteracting the effects of pouching derived from insufficient management.

Enhanced conservation of fish assemblages and the surrounding ecosystem in the eastern part of Los Canarreos archipelago would contribute to preserve the goods and services delivered by coral reefs and will be a significant contribution to the marine conservation of these ecosystems in Cuba and the Caribbean Sea. We recommend immediate actions within a strategy of precautionary management including, but not

limited to, the appointment of staff for the administration of Cayo Largo MPA, frequent monitoring and effective enforcement. Improving the environmental quality of these coral reefs under effective management schemes will be instrumental for keep running touristic activities such as SCUBA diving and recreational fishing. Future studies should focus on the effects of habitats features (*e.g.*, complexity, diversity) and surrounding coastal ecosystems (*e.g.*, Zapata swamp) on coral reef fish assemblages.

## ACKNOWLEDGEMENTS

We acknowledge CIM-UH's staff for assistance with field work. We also thanks to R/V Felipe Poey crew for the support during the sampling. Thanks to the Woods Hole Oceanographic Institution and the Dalio Foundation through the Dalio Ocean Initiative, for helping to establish this U.S.-Cuban research partnership. Thanks to Operation Wallacea and Jorge Angulo for allowing the acquisition and use of stereo-DOV in Cuba. The authors thank to Eric Ward, (Academic Editor of PeerJ), Susana Perera, Ariagna Lara and one anonymous reviewer, who improved the manuscript.

### Funding

Woods Hole Oceanographic Institution and the Dalio Foundation through the Dalio Ocean Initiative supported the project. The funders had no role in study design, data collection and analysis, decision to publish, or preparation of the manuscript.

### Grant Disclosures

The following grant information was disclosed by the authors:
Woods Hole Oceanographic Institution.
Dalio Foundation.

### Competing Interests

The authors declare that they have no competing interests.

### Author Contributions

- Zenaida María Navarro-Martínez conceived and designed the experiments, analyzed the data, prepared figures and/or tables, authored or reviewed drafts of the article, and approved the final draft.
- Maickel Armenteros conceived and designed the experiments, analyzed the data, prepared figures and/or tables, authored or reviewed drafts of the article, and approved the final draft.
- Leonardo Espinosa conceived and designed the experiments, authored or reviewed drafts of the article, and approved the final draft.
- Patricia González-Díaz conceived and designed the experiments, analyzed the data, authored or reviewed drafts of the article, and approved the final draft.
- Amy Apprill conceived and designed the experiments, analyzed the data, authored or reviewed drafts of the article, and approved the final draft.

## Field Study Permissions

The following information was supplied relating to field study approvals (*i.e.*, approving body and any reference numbers):

Field surveys were conducted under the permission No. 2015/25 for accessing to natural and mountainous areas, emitted by the Ministerio de Ciencia, Tecnología y Medio Ambiente de Cuba (CITMA), in favor to the Centro de Investigaciones Marinas, Universidad de La Habana.

## Data Availability

The raw data is available in the Supplemental Files.

## Supplemental Information

Supplemental information for this article can be found online at http://dx.doi.org/10.7717/peerj.14229#supplemental-information.

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
