# Peer review of "Coral reef fish assemblages exhibit signs of depletion in two protected areas from the eastern of Los Canarreos archipelago (Cuba, Caribbean Sea)"

_PeerJ, doi:10.7717/peerj.14229_

## Round 0.1 · original submission · Minor Revisions

This paper has now been seen by 3 reviewers, and all recommend a minor revision. Please pay attention to detail throughout - consider changing the title, and add more details to the discussion.

Reviewer 1 ·

Basic reporting

The article is well written, with adequate and understandable English language. The literature used is up-to-date and consistent with the research topic, and in my annotations, I suggest incorporating some more. It is a professional article with correct tables and figures. The results achieved are relevant.

Experimental design

It is an original article and the data is unpublished. The research question is well defined. I recommended to review the objectives set out in the document. The investigation is rigorous using metodology and correct statistical analisys. I also make some suggestions in this regard.

Validity of the findings

Article with impact and novelty. Its replication is encouraged and its publication is justified. All underlying data has been provided; they are robust, statistically solid and controlled, taking into account the sampling frequency that was carried out.
It is recommended to review the conclusions according to the type of result achieved. They are linked to the research question.

Additional comments

I thank the authors for their effort and for sharing the results of their research with the scientific community. I recommend reviewing the remarks and suggestions that I make in the attached document.

Annotated reviews are not available for download in order to protect the identity of reviewers who chose to remain anonymous.

·

Basic reporting

no comment

Experimental design

no comment

Validity of the findings

I suggest making more evident in the discussion, the difference between CCCR and CL in terms of management and enforcement, keeping in mind that CCCR is a legally approved MPA (although with many limitations) and CL has been identified for its biodiversity values as a possible MPA, but so far, as you duly clarify, unfortunately, it does not have administrative staff and does not have legal approval.

From my point of view, the study compares an approved MPA (with limited enforcement), a proposed MPA (unstaffed and with no enforcement), and a non-MPA zone, but I suggest not to conclude in the case of CL that the MPA fails.

I would also suggest addressing this comment in the title of the article.

I suggest including in the discussion information that analyzes possible causes of the results of the indicators, for example in CL, perhaps greater access to the reefs, because as mentioned in M&M, in this region there are hotels and marinas that probably give access to potential illegal fishermen, or where diving activities are frequently carried out, etc.

Additional comments

I believe that a topic of great interest is being addressed, taking into account that these are poorly studied areas and the implications of the work for the conservation of coral reefs in the Southwest region of Cuba and the Caribbean.

The manuscript is very well written and referenced.

After attending to the brief comments, I consider the article suitable for publication in PeerJ.

·

Basic reporting

The figures are relevant to the content of the article, with sufficient resolution and appropriately described. However, I suggest in Figure 1 to highlight the sampling sites that are outside the protected areas.
The literature used is current and appropriate for the research. However, bibliographic citations must be homogenized, using ¨and¨ or ¨&¨.

Experimental design

The materials and methods are described with sufficient information to be reproducible by another investigator, however, I suggest explaining more why the number of transects used is not the same in all the sites.

Validity of the findings

This research contributes significantly to the knowledge of the ichthyofauna in the Los Canarreo archipelago, Cuba. It provides valuable information for the management of protected areas. However, I suggest reviewing some aspects to improve the understanding of this article:

1. According to the results shown, I agree with the authors regarding the apparent little effect of the two protected areas studied, but there are aspects that are not clear to make this statement:
• What was the ichthyofauna like before the creation of these protected areas? The information is available in the study area, but with other sampling methodologies.
• Sites outside protected areas are very close, could there be an export effect?
• The authors affirm that Cayo Largo is an area that does not have personnel for its administration and management, so how can we expect a positive effect from the area if there is no control?

2. Due to the last questions, I suggest changing the title of the post to not be categorical if there are not enough elements to show that the areas have not had an effect as protected areas.

---

## Round 0.2 · accepted · Accept

Thanks for addressing reviewers comments during the previous round of revision, you have incorporated all of their feedback and suggestions, and I appreciate the point by point rebuttal.